# Multi-Authority Revocable Access Control Method Based on CP-ABE in NDN

**Zhijun Wu *****, Yun Zhang and Enzhong Xu**

School of Electronic Information and Automation, Civil Aviation University of China, Tianjin 300300, China; chunyyzhang@163.com (Y.Z.); xez0120@163.com (E.X.)

**\*** Correspondence: zjwu@cauc.edu.cn; Tel.: +86-022-24092827

**Abstract:** For the future of the Internet, because information-centric network (ICN) have natural advantages in terms of content distribution, mobility, and security, it is regarded as a potential solution, or even the key, to solve many current problems. Named Data Network (NDN) is one of the research projects initiated by the United States for network architecture. NDN is a more popular project than ICN. The information cache in the NDN separates content from content publishers, but content security is threatened because of the lack of security controls. Therefore, a multi-authority revocable access control method based on CP-ABE needs to be proposed. This method constructs a proxy-assisted access control scheme, which can implement effective data access control in NDN networks, and the scheme has high security. Because of the partial decryption on the NDN node, the decryption burden of the consumer client in the solution is reduced, and effective user and attribute revocation is achieved, and forward security and backward security are ensured, and collusion attacks are prevented. Finally, through the other security and performance analysis of the scheme of this paper, it proves that the scheme is safe and efficient.

**Keywords:** Information-Centric Network (ICN); Naming Data Network (NDN); access control; multiple Authorization; Ciphertext-Policy Attribute-Based Encryption (CP-ABE)

---

## 1. Introduction

With the continuous emergence of big data, artificial intelligence, Internet of Things Plus, cloud computing, and other technologies, there are a variety of content-oriented networking applications on the Internet [1,2]. As long as the content is secure and reliable, users are increasingly interested in using and accessing web content, and users do not care who sends the content. The increasing use of peer-to-peer (P2P) in content delivery networks (CDN) proves this situation [3]. As content distribution increases, the host-centric model-based Internet architecture is difficult to integrate new consumption methods and mobility needs. This situation has inspired experts to create other methods to disseminate content [4,5]. Therefore, the researchers created an Information Center Network (ICN). ICN connects content consumers directly with content by content name. In this way, the content consumer does not need to identify the address of the content owner, but only the reliable copy of the content. Content owners publish content and store it on the network [6]. This design enables content to be delivered to consumers effectively. There is an advantage in retrieving content and content sharing and distribution, but because content and content publishers are decoupled, data owners cannot protect the privacy and security of data as much as managing local data. Traditional public key cryptosystems only support "one-to-one" encryption and lack flexible access control strategies. This poses a huge challenge to security issues during content caching and retrieval. Therefore, a mechanism to ensure data confidentiality should be adopted [7]. To ensure data security, Named Data Network (NDN) networks need to introduce access control methods. However, in the NDN cache

system, the NDN routing node cannot be completely trusted, because the NDN routing node may conspire with other users to decrypt the content of the data. In order to realize the secure and effective access control effect on NDN network data, we adopt a ciphertext policy attribute-based encryption method (CP-ABE). CP-ABE has the characteristics of flexibility, one-to-many, rich attribute expression, and small granularity in encryption, which is suitable for content distribution and sharing in the NDN environment. The existing ciphertext-based attribute-based encryption scheme is only applicable to the case where the distribution of user attributes and keys belongs to the same organization, but the user attributes may be stored in different cloud storage servers. In the actual application environment, attribute revocation, user permission change, etc., may occur, so implementing data security sharing and implementing attribute revocation on multiple servers becomes an urgent problem to be solved.

Our contributions can be summarized as follows.

We propose a CP-ABE-based multi-authority revocable access control method. This method constructs a proxy-assisted access control scheme to achieve secure and effective access control of data in the NDN network. We design an effective user and attribute revocation method, and ensure forward security and backward security to prevent collusion attacks. In addition, from a comparative analysis of performance and safety, we show that our scheme is safer and more efficient than other schemes.

## 2. Related Works

The information caching technology in NDN simplifies the request and distribution of content, which is beneficial to solve the problem of efficient access and secure transmission of data in cloud computing. The information cache decouples the publisher from the data, causing the publisher to lose control of the content and the risk of data leakage. Therefore, access control is especially important for NDN security. In order to protect data privacy and realize effective access control, Shamir first proposed identity-based encryption in 1984 [8]. On this basis, Sahai and Waters first proposed fuzzy identity encryption, also known as attribute encryption [9]. After that, Goyal et al. [10] and Bethencourt et al. [11] constructed two more practical attribute encryption schemes, namely Key-Policy Attribute-based Encryption (KP-ABE) and Ciphertext-Policy Attribute-based Encryption (CP-ABE). In the original attribute encryption scheme, only a single central authority manages all attributes. In real life, it is not practical to issue a key for all attributes of a user, the authority has a heavy workload and requires complete credibility, and a single authorization failure can cause a system crash. For the purpose of solving the above problems, the attribute encryption schemes of multiple multi-attribute authorities are proposed in the subsequent research. The idea of managing different attributes of users by multiple attribute authority (AA) is first proposed by Chase et al. in [12]. Managing multiple attributes by multiple attribute authorities can alleviate the workload of a single authority, while avoiding system crashes that can result from a single authorization failure.

CP-ABE is considered to be one of the most fitness technologies for data access control in cloud storage. At present, CP-ABE is divided into single-privilege CP-ABE and multi-privilege CP-ABE program [13]. Hur et al. [14] proposed another CP-ABE scheme with fine-grained attribute revocation. They re-encrypt the ciphertext using the attribute group key. However, their method is not effective in resisting collusion attacks, and may even fail to resist. Yang et al. [15,16] proposed two CP-ABE schemes for multi-authorization centers that support decryption outsourcing, enabling effective cloud data sharing and user revocation. Kai et al. [17] proposed a smart city cloud data agent auxiliary access control scheme to realize access control and user revocation of smart city cloud data. Both NDN information caching and cloud storage have problems with the decoupling of content and content owners. In order to realize secure and efficient access control of NDN cache content, attribute-based access control in NDN has attracted more and more researchers' attention. Single authorization: Li et al. designed a CP-ABE-based naming scheme [18] that increases the naming overhead and does not consider the issue of consumer revoke. In terms of content caching, S1iva et al. proposed an instant-revokable NDN access control mechanism, which solves the user-level revocation problem, but does not consider the attribute-level revocation and key leakage problems. The proxy server performs user authentication online at all times, and no specific formalization algorithm and

security analysis and certification are given in the text. Multi-authorization: Tao et al. proposed a novel NDN access control scheme to solve this problem [19]. The system is based on CP-ABE multi-authorization, revocable access control scheme. The solution implements an indirect revocation of consumer users. In addition, the program did not use experiments to prove the time of content release and content request.

However, as far as the NDN system is concerned, attribute revocation and users have been its challenging problems. There are many reasons why attribute revocation and users will cause challenges in the NDN system, such as the following reasons. The first reason is that because of the content caching mechanism of the NDN, even after the content is successfully forwarded, it cannot be reused by other users, and the cached content in the content store in the NDN can be reused. The second reason is that since multiple users may share each attribute, the revocation of any one attribute may have an essential impact on other users in the attribute group. The third reason is that many solutions do not support attribute revocation and flexible user in a multi-authority NDN cache system. Therefore, this paper proposes a multi-authority revocable access control method based on CP-ABE in NDN. This method realizes data access control and user revocation and attribute revocation of rights in NDN cache system through proxy decryption, and can resist collusion attack. Forward security and backward security are also achieved. Compared with single-privileged CP-ABE scheme and multi-authority CP-ABE scheme, the attributes of the multi- authority CP-ABE scheme come from different attribute domains and are authorized by different authorization centers. Therefore, the multi- authority CP-ABE scheme is more appropriate for content access control of the NDN cache system.

## 3. Design and Implementation of Access Control Method

Based on the Kai Fan scheme [17], we propose a CP-ABE-based multi-authority revocable access control method that is different from the traditional single-authorization center scheme to enable fine-grained access control of cached data. In the multi-authorization center attribute-based encryption mechanism, the user's attributes come from multiple attribute domains that are jointly managed by different authorization centers. This approach can greatly reduce the administrative burden of a single authorization center and can improve the system's ability to resist collusion attacks.

### 3.1. Overall Design of Access Control Method

The CP-ABE-based access control system in the NDN proposed in this paper mainly includes the following five parts: NDN cache node, global certificate authority (CA), attribute authorization center (AAs), content publisher, and content consumer. Figure 1 demonstrates flow of the NDN access control system.

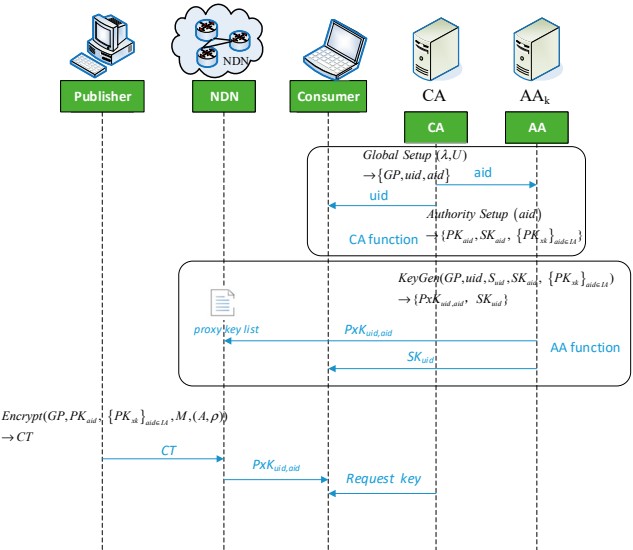

**Figure 1.** Flow of the Named Data Network (NDN) access control system.

1.    Central Authorization Center (CA)

The CA is fully trusted. It is primarily responsible for the authentication of the Authorization Center and consumer users in the system, and publishes a globally unique identifier uid for each NDN consumer user, giving each AA a globally unique identity $aid_i$, which does not participate in any work related to property management and key generation.

2.    Attribute Authorization Centers (Hereinafter abbreviated as AA or $AA_i, i = 1, \ldots, k$)

Each $AA_i$ is independent of others and each $AA_i$ manages different attributes and generates related keys. In this scenario, each $AA_i$ can control any number of attributes, and one $AA_i$ can only manage each attribute. Each $AA_i$ is responsible for generating an attribute public key PK$x$ for the attributes of the management attribute domain, and dividing the consumer key into a proxy key $PxK_{uid,aidi}$. $PxK_{uid,aidi}$ will be sent to the NDN routing node for storage, and the routing node will add the consumer user's $PxK_{uid,aidi}$ to the proxy key list $LPxK$. The user secret key $SK_{uid}$ is sent to the corresponding consumer users that are kept by the consumer users themselves.

3.    Content Publisher

The content publisher defines an access control policy for attributes from $AA_i$, and then encrypts the content that needs to be shared. These encrypted content ciphertexts are cached at the NDN routing node.

4.    NDN cache nodes

They can provide storage services for NDN network content through their own storage policies, and provide response packets for them according to the consumer's interest package. When a user issues a file query request, the cloud service provider first checks if his set of attributes conforms to the access structure. If his attribute meets the access structure, then the partially decrypted ciphertext is computed. Only legitimate consumer users can decrypt the content of the ciphertext by using the proxy key through the NDN routing node. The legitimate consumer users then send the decrypted data to the appropriate consumer.

5.    Content Consumer

Content consumers can obtain encrypted data from their neighbor routers or publishers while requesting their keys from the relevant CA. The NDN routing node partially decrypts it using its proxy key. After the consumer user obtains the partially decrypted content, the user can use the key to decrypt the remaining portion to obtain the plaintext data.

The main steps of the mechanism are divided as follows. The first step is for the CA to initialize the system to issue a globally unique identity to the AA and to issue a globally unique identity to the content consumer. The second step AA is responsible for managing the attributes of the attribute domain and generating the attribute public key. In addition, the AA will generate a proxy key and add it to the NDN routing node, and the AA will send the private key to the corresponding consumer user. In the third step, the publisher needs to use the public key to encrypt the information to be published. The fourth step is that the consumer user requests a key from the CA or uses the proxy key of the NDN node for partial decryption.

### 3.2. Access Control Method Implementation

#### 3.2.1. System Initialization

The consumer needs to obtain the access permission of the content to obtain the content when the user first visits, and the consumer user issues the signed interest package. The format of the interest package name can be "ndn/user/number/i/registration," and the registration is completed and the permission is obtained. In the process, the content publisher obtains the consumer user attribute and also authenticates the user's identity through the attribute. The definition and retrieval of attributes is transparent to NDN. After receiving the interest package requesting the content, the content publisher verifies the integrity and authenticity of the registered interest package. If the interest package is authenticated, the content publisher will send a registration confirmation packet to the consumer, and the publisher will verify the user and complete the user registration process (see Figure 2). The initialization process needs to execute the following functions:

1. Global Setup $(\lambda, U) \rightarrow \{GP, uid, aid\}$

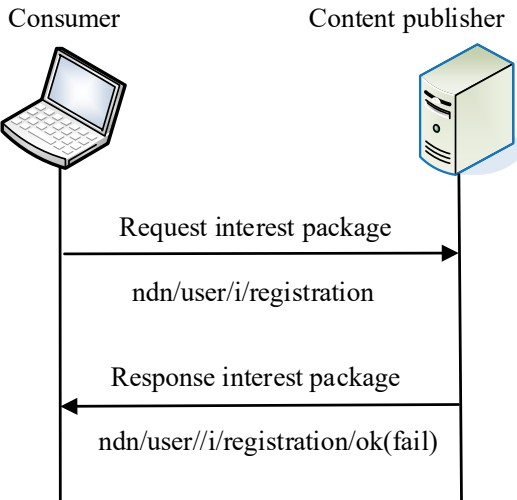

**Figure 2.** User registration process.

Global initialization function, CA first selects a system security parameter $\lambda$ and an attribute domain $U$, and finally generates a global public parameter $GP$, $AA_i$ identity *aid* and user identity *uid*.

2. Authority Setup $(aid) \rightarrow \{PK_{aid}, SK_{aid}, \{PK_{xk}\}_{aid \in IA}\}$

$AA_i$ initialization function that is executed by each $AA_i$. Enter aid, output a pair of $AA_i$ public and private keys $PK_{aid}$ and $SK_{aid}$, and generate an attribute public key $\{PK_{xk}\}_{aid \in IA}$ for each attribute managed by the $AA_i$.

### 3.2.2. Content Publishing Process

Content publishing is the basis of NDN data sharing. Before publishing content, the content is first encrypted by CP-ABE algorithm to obtain ciphertext CT. The content publishing process is shown in Figure 3. The content publishing process needs to execute the following functions:

3. Encrypt $(GP, PK_{aid}, \{PK_{xk}\}_{aid \in IA}, M, (A, \rho)) \rightarrow CT$

The encryption function is executed by the data owner. Input $GP$, $PK_{aid}$ and $\{PK_{xk}\}_{aid \in IA}$, plaintext $M$ and access structure $A$, and output ciphertext $CT$.

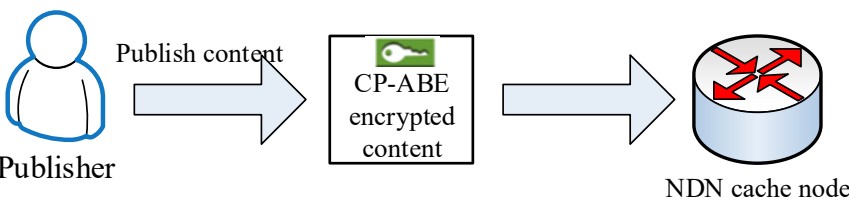

**Figure 3.** Content release process.

4. KeyGen $(GP, uid, S_{uid,aid}, SK_{aid}, \{PK_{xk}\}_{aidIA}) \rightarrow \left(PxK_{uid,aid}, SK_{uid}\right)$

AA executes the key generation function. Enter $GP$, $uid$, a set of consumer user attribute sets $S_{uid,aid}$, $SK_{aid}$ and $\{PK_{xk}\}_{aidIA}$, output proxy key $PxK_{uid,aid}$ and user private key $SK_{uid}$.

The encrypted content ciphertext is published to the NDN network as part of the packet data. This packet can be cached by any router and accessed through the NDN infrastructure. This part is usually massive data, so consumers can get the content of interest from the neighbor router that increases distribution efficiency and efficiency of data transmission.

### 3.2.3. Content Request Process

After the content is published to the NDN network, in order to enable the consumer to access the content of interest, two steps are required, first the proxy decryption of the NDN cache node and the user decryption of the consumer client. The content request process needs to execute the following functions:

5. PxDecry $(GP, CT, PxK_{uid,aid}, \{PK_{xk}\}_{aid \in IA}) \rightarrow CT'$

The NDN routing node performs a proxy decryption algorithm. Input $GP$, $CT$, $PxK_{uid,aid,}$ and $\{PK_{xk}\}_{aid \in IA}$, and the partially decrypted ciphertext $CT'$ is output.

6. Udecrypt $(CT, CT', SK_{uid}) \rightarrow M)$

Consumer user decryption function is executed by the consumer. Input $CT$, $CT'$, and $SK_{uid}$, and output plaintext $M$.

### 3.3. Revocation of Consumer Privilege

As long as the consumer user is restricted from accessing the data file again, a user revocation is performed. In this method, the consumer user does not need to update the keys of other users who have not been revoked, they only need to re-encrypt the ciphertext. The content publisher only needs to issue an undo message containing the revoked user's identity *uid* to NDN routing node, and the routing node deletes the revoked user's proxy key $PxK_{uid,aid}$. Once $PxK_{uid,aid}$ is removed, NDN

router can no longer execute the *ProxyDec* proxy decryption algorithm for the revoked user. Therefore, revoked user cannot perform decryption process. The undo process requires the execution of URev $(uid, L_{PxK}) \rightarrow L\prime_{PxK}$. The user revocation function is executed by the NDN routing node, inputting the *uid* of the consumer user and the proxy key list $L_{PxK}$, and outputting the updated proxy key list $L\prime_{PxK}$.

Consumer Attribute Revocation

For attribute revocation, this method introduces the idea of the attribute version number. When the authorization center is initialized, first select a version number for each attribute in system. When attribute revocation happens, simply update the affected part of the ciphertext and key. The affected authorization center will select a new attribute version number for the revoked attribute and generate a relevant new key to update the affected ciphertext. Consumer property revocation involves two processes: key update and ciphertext re-encryption.

7. The key update

Key update needs to execute the ReKeyUpdate function. The relevant authorization center first produces a novel attribute version number, then changes attribute public key for revoked attribute, and broadcasts a message to the data owner in the system, making them to receive updated attribute public keys. At the same time, the NDN routing node updates the proxy key using the proxy key update function *PxKUp* date. The consumer key update needs to perform the following functions:

- ReKeyUpdate $\left(uid, PxK_{uid,aid}, v_{xk}\right) \rightarrow \{VUK_{Xk}, PxUK_{Xk}\}$

$AA_i$ executes this key material update function. Enter the *uid*, $PxK_{uid,aid}$ of the consumer user that has not been revoked, and the current attribute version number $v_{xk}$, and output the version update key $VUK_{Xk}$ and the agent update key $PxUK_{Xk}$.

- PxKUpdate $\left(uid, PxK_{uid,aid}, PxUK_{Xk}\right) \rightarrow PxK^*_{uid,aid}$

The NDN routing node performs the proxy key update function. The function inputs the *uid*, $PxK_{uid,aid}$ of the consumer user that has not been revoked, and the current attribute version number $v_{xk}$. output version update key $VUK_{Xk}$ and agent update key $PxUK_{Xk}$.

8. The ciphertext re-encrypted

Data owner uses the version update key to calculate the ciphertext update key through the ciphertext material update function CTUpdate and sends it to the NDN cache node, and the NDN cache node performs the ciphertext re-encryption function ReEnc to perform the ciphertext re-encryption. Consumer ciphertext re-encryption needs to perform the following functions:

- CTUpdate $(VUK_{xk}, CT) \rightarrow CUK_{xk}$

The ciphertext material update function is executed by the content publisher and input $VUK_{xk}$, $CT$ and output the ciphertext update key $CUK_{xk}$.

- ReEnc $(CT, CUK_{xk}) \rightarrow CT*$

The NDN routing node performs the ciphertext re-encryption function. Enter the current ciphertext $CT$ and $CUK_{xk}$.

*3.4. NDN Router*

The NDN router can change the NDN cache node to aggregate proxy re-encryption. The forwarding operation of the data packet and the interest packet by the NDN route cache node is based on the name routing and forwarding. The content name is an opaque binary code sequence for the

router. When forwarding the data packet or the interest packet, the name and the information in the FIB need to be processed to match. For example, /university/computer/access control, when it matches the interest packet that needs to perform proxy re-encryption service, it will complete the corresponding operation through the agent module to achieve the purpose of the consumer requesting data, complete efficient data distribution, and optimize the network system performance. Figure 4 illustrates the relationship of the NDN cache node and its main components in the access control method.

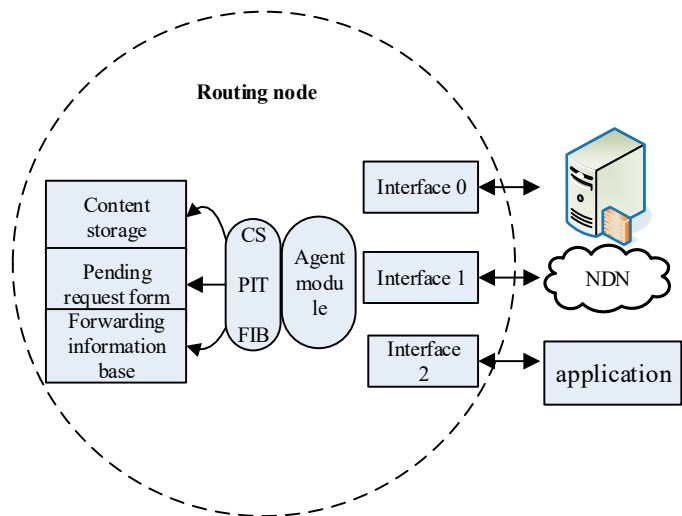

**Figure 4.** Relationship between NDN routers and proxy servers and applications.

## 4. Security Analysis

In this section, the security analysis of the CP-ABE-based multi-authorization revocable access control method will be introduced, including data confidentiality, forward security and post-security, and collusion attacks by multiple consumers.

### 4.1. Confidentiality of Data

When the user's attributes meet the requirements of the access structure, it can use the key to implement the decryption process of the data. If the user's attributes cannot meet the requirements of the ciphertext access structure, then they cannot accept the ciphertext information during the decryption proxy process. Since the proxy decryption process can be decrypted by the cloud private key, the lack of the user agent key will not help the data decryption process. When a user is revoked, the cloud server must delete the proxy key it holds. When the proxy key is deleted and does not exist, user will not be able to accept the partially decrypted ciphertext.

### 4.2. Forward Security and Post Security

This paper proposes an access control scheme based on CP-ABE, which can guarantee the forward and backward security of the NDN cached data for newly added and revoked consumers. Forward security denotes the revoked user unable decrypt novel ciphertext that needs to be revoked for decryption. Backward security denotes newly added users can also decrypt previously ciphertexts. Forward security denotes the revoked user cannot decrypt the new content of the ciphertext that demands to be revoked for decryption. Backward security denotes newly added users can also decrypt previously public the content of the ciphertext, and if it has enough attributes, it uses the previous public key encryption. When the consumer is revoked, the NDN cache node will delete its proxy key. Therefore, user who is revoked is unable to decrypt the ciphertext by utilizing its proxy key to ensure the forward security of the system's cached data. When a whole new consumer adds system, relevant

ciphertext information is re-encrypted so that it can also decrypt the ciphertext, ensuring backward security of the cached data.

*4.3. Collusion Attack*

In traditional attribute encryption schemes, a single AA combines parts of a user's private key (corresponding to different attributes) through key randomization. This randomization makes the elements of different keys of one user compatible with each other, but cannot be combined with the key elements of other users. This key randomization technique is not suitable for decentralized attribute encryption, because no CA can combine these elements, and each key comes from a different AA. In order to overcome this difficulty, this article uses different global identities to combine different user key parts to meet the needs of anti-collusion. Consumer users base this solution on the CP-ABE access control method to defend against conspiracy attacks. Suppose the number of permissions involved in ciphertext is $n$, and the number of collusion permissions is $m$. If $m = n$, intuitively, these permissions can get all the keys that can be used to decrypt the content of ciphertext. If $m \leq n - 1$, there is at least one privilege that the key cannot be obtained. Therefore, ciphertext cannot be decrypted. This scheme implements a collusion attack of up to $(n-1)$ privilege.

## 5. Analysis of Performance

We performed the analysis of the multi- authority CP-ABE access control method in this section. The content of the analysis is expanded from three aspects: flexibility analysis, calculation overhead, and efficiency analysis.

*5.1. Flexibility Analysis*

Table 1 compares the CP-ABE schemes previously studied by relevant experts from the four aspects of access structure, type of authorization center, ability to resist collusion attacks, and authority revocation.

**Table 1.** Flexibility comparison of the program.

| Program | Access Structure | Authorization Center Type | Can Resist Collusion Attacks | Authority Revocation |
|---------|------------------|---------------------------|------------------------------|----------------------|
| Tao [19] | LSSS | Multiple authorization center | Yes | Indirect revocation |
| Silva [5] | Tree structure | Single authorization center | Yes | User revocation |
| Lewko [20] | LSSS | Multiple authorization center | Yes | No |
| Li [18] | Tree structure | Single authorization center | Yes | No |
| This article | LSSS | Multiple authorization center | Yes | Direct revocation |

As can be seen from Table 1, this scheme is more flexible than other solutions. Because it adopts the LSSS access structure, the scheme supports fine-grained access control, supports multiple authority, can resist collusion attacks, and supports user and attribute revocation. These features make the solution proposed in this chapter more suitable for practical application requirements.

*5.2. Calculation Overhead*

In order to analyze the computational cost of the scheme, the proposed scheme is compared with the existing scheme. $T_e$ and $T_m$ are one bilinear calculation time and one exponentiation time. $l$ is the number of attributes, $N_A$ and $M$ are the number of authorities and the size of the encrypted file. $|G_T|$ and $|G|$ are the size of $G_T$ and $G$. Let $IA$ denote the universe set of all AA and each AA manages $xk$ ($xk(k \in IA)$) attributes. For a user with uid, let $xk, uid$ denote the number of attributes obtained from

the authority with $aid_k(k \in IA)$. The calculation overhead results of each scheme are shown in Table 2. The analysis includes ciphertext size, user encryption time, and user decryption time.

**Table 2.** Calculation cost comparison.

| Program | Secret Key | Ciphertext Size | Consumer Encryption Time | Consumer Decryption Time |
|---|---|---|---|---|
| Yang[15] | $|G| + \sum_{k \in IA} xk, uid|G|$ | $N_A|G_T| + (4l+2)|G| + C$ | $N_A T_e + (4l + N_A + 2)T_m$ | $(4l+2)N_A T_e + lN_A T_m$ |
| Qian [21] | $|G| + xk, uid|G|$ | $N_A|G_T| + (l+1)|G| + C$ | $(N_A + 1)T_e + (l+1)T_m$ | $(l+1)T_m$ |
| Wu [22] | $|G| + 2(xk, uid)|G|$ | $N_A|G_T| + (2l+1)|G| + C$ | $N_A T_e + (3l + +N_A + 1)T_m$ | $(l + N_A + 2)T_m$ |
| This scheme | $|G| + 2(xk, uid)|G|$ | $N_A|G_T| + (2l+1)|G| + C$ | $N_A T_e + (3l + N_A + 1)T_m$ | $T_m$ |

It can be seen from the results in Table 2 that the ciphertext length of the scheme in this chapter is slightly shorter than the other scheme. the encryption time of the consumer user is also slightly shorter than the other scheme. The user decrypts time because part of calculation is transferred to the NDN cache node in decryption phase. It is significantly shorter than the other. In summary, the computational overhead of this chapter's scheme is significantly better than the other scheme.

*5.3. Efficiency Analysis*

In order to implement the simulation experiment using the tool Charm, a framework for quickly implementing cryptography schemes and protocols is required. The LSSS scheme required in the attribute-based encryption system is also provided, and the PBC library is used to implement the main group operation. The simulation experiment environment is on Intel Xeon 3.60 GHz processor W-2133, 16 GB memory, install VMware Workstation 10.0 virtual machine, and install Ubuntu 16.04 system in virtual machine.

As can be seen from the Figure 5, with the increase of the number of attributes, our scheme has less encryption time than Lewko's scheme [20] and Fan's scheme [17]. In addition, the encryption time increases with the number of authorized centers, and increases almost linearly. The length of C in ciphertext is related to the number of attributes. The more attributes, the longer C, so the longer the encryption takes.

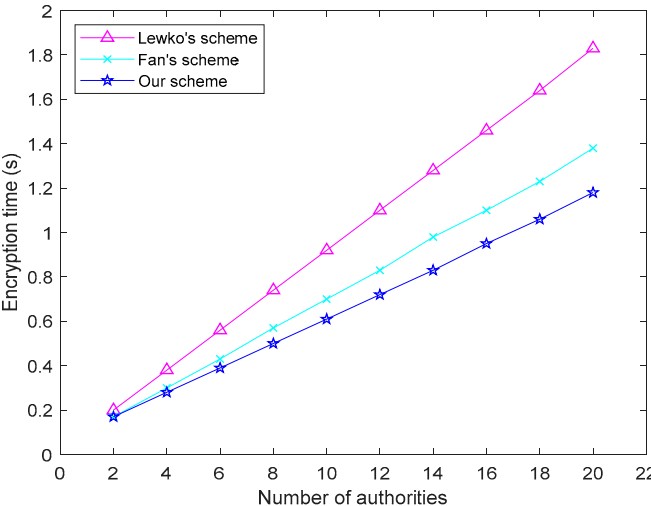

**Figure 5.** Relation between consumer decryption and encryption time and number of authorities.

As can be seen from the Figure 6, the consumer user decryption time of the solution in this paper is almost constant, because the NDN cache node completes part of the calculation in the decryption

process, and the obtained partially decrypted ciphertext is sent to the consumer user. Decryption part shows that the user can restore the plaintext only by completing part of the calculation.

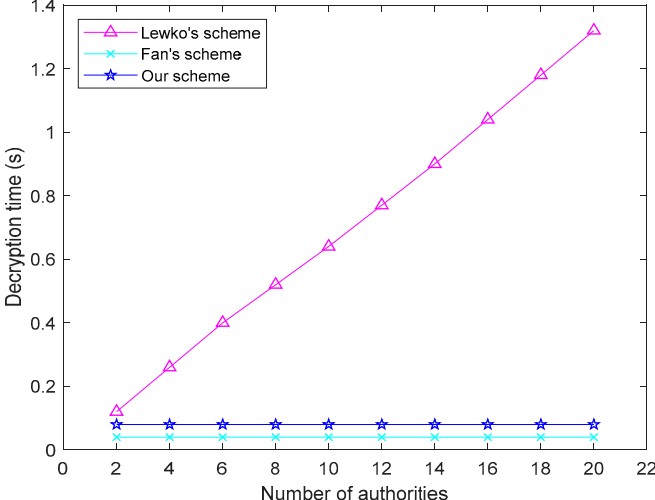

**Figure 6.** Relation between consumer decryption time and number of authorities.

Figure 7 shows the result of proxy decryption for the NDN routing node. It can be seen from Figure 7 that our scheme has less proxy decryption time than the Fan scheme [17] as the number of attributes increases. It can be seen that the time of proxy decryption increases with the number of authorities, and it increases almost linearly. In formula (4–7) of [13], $K_{uid, x}$ is related to the number of attributes, the greater the number of attributes, the longer its length, and therefore, the longer the agent decrypts.

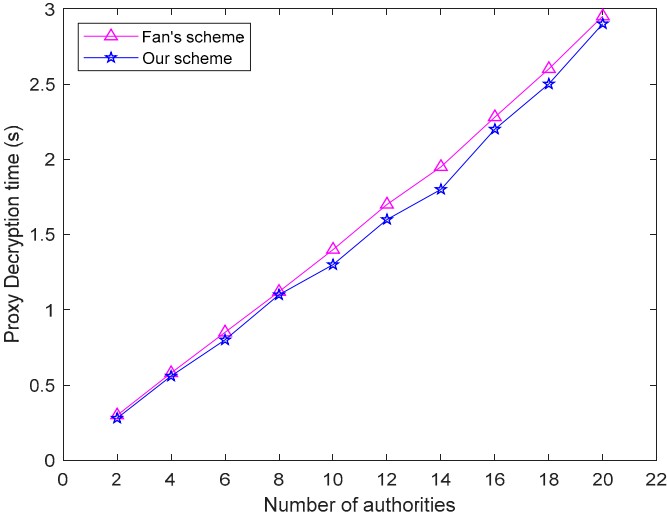

**Figure 7.** Relation between proxy decryption time and number of authorities.

Figure 8 shows the relation between the time consumed by ciphertext re-encryption and the number of revoked attributes in the attribute revocation phase. Obviously, the more the number of attributes that are revoked, the longer the ciphertext re-encryption consumes and the almost linear growth. In this scenario, there is no need to update all ciphertexts when attribute revocation occurs, just update those ciphertexts associated with the revoked attributes.

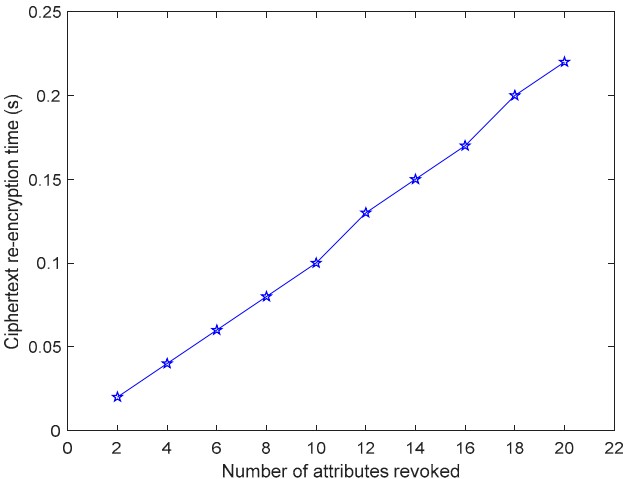

**Figure 8.** Time spent on ciphertext re-encryption.

## 6. Summary

Information caching technology in named data networks improves the efficiency of data distribution. However, information caching decouples data publishers from data, causing data in storage nodes to face threats issues because of the lack of security controls. The security issues of NDN content affect NDN applications and deployment. For the purpose of solving the access control problem of cached content in the named data network NDN, this paper constructs a proxy-assisted access control scheme. The multi-authorization revocable access control method based on CP-ABE implements secure and effective access control of data in the NDN network. Because of the partial decryption on the NDN node, the decryption burden of the consumer client in the solution is reduced, and the method implements effective user and attribute revocation, and ensures forward security and backward security against collusion attacks. Finally, the performance and security analysis of the scheme of this paper proves that the scheme is safe and efficient.

**Author Contributions:** Z.W. and E.X. contributed to the conception of the study, Y.Z. collected important background information, performed the experimental analyses and wrote the manuscript. All authors read and approved the final manuscript.

**Funding:** This work was supported in part by the joint funds of National Natural Science Foundation of China and Civil Aviation Administration of China (U1933108), the Key Program of Natural Science Foundation of Tianjin (17JCZDJC30900), the Scientific Research Project of Tianjin Municipal Education Commission (2019KJ117), and the Fundamental Research Funds for the Central Universities of China (3122018D34007, 3122018C003).

**Conflicts of Interest:** The authors declare no conflict of interest.

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
