# Peer review of "Multi-Authority Revocable Access Control Method Based on CP-ABE in NDN"

_futureinternet, doi:10.3390/fi12010015_

Round 1
Reviewer 1 Report
This paper presents a mechanism to perform access control to the contents distributed by Named Data Networks. The access control mechanism also supports the user's authorization revocation. Ensure forward and backward security and resistance to collusion attacks are the main objectives of the presented mechanism. A mechanism with these characteristics is critical to allow the Named Data Network massification.
A deep and careful revision is required. There are many sentences without meaning or not precise. Just a few examples:
Line 35 ..."In order meet the needs of the future internet, ICN replaces the content name (address) with the content name and is considered to be the main element in the architecture."
Line 97 ..."The first reason is that due to NDN cache system have many users, users may change frequently."
Figure 1 doesn't help the reader to understand how the proposed mechanism works. Probably some information can be removed. A sequence diagram presenting the main tasks should be considered.
The authors should also start giving an overview of the main steps of the mechanism.
How the proposed mechanism avoids, the collusion attacks must be better explained.
As a conclusion, NDN stills a hot topic with a lot of open issued to be solved. The problem identification is correct; however, the solution details are not clearly explained. An in-depth revision of the language and syntax style is required, The content must also be improved.
Author Response
Dear Editor of the Journal of Future Internet:
Thank you for your professional review and hard work! Thank you for your valuable comments and Suggestions! These Suggestions and opinions on our paper writing and research work are of great guiding significance! At the same time, we need to urge more serious attitude in scientific research and learning!
According to the opinions of the reviewers, we replied below.
Responds to the reviewer’s comments:
To Reviewer 1:
1. Response to comment: Line 35 ..."In order meet the needs of the future internet, ICN replaces the content name (address) with the content name and is considered to be the main element in the architecture."
Response:
Thanks to the reviewers for their comments, we have replaced the previous sentence with the following.
ICN connects content consumers directly with content by content name. In this way, the content consumer does not need to identify the address of the content owner, but only the reliable copy of the content.
2. Response to comment: Line 97 ..."The first reason is that due to NDN cache system have many users, users may change frequently."
Thanks to the reviewers for their comments, we have replaced the previous sentence with the following.
The first reason is that due to the content caching mechanism of the NDN, even after the content is successfully forwarded, it cannot be reused by other users, and the cached content in the content store in the NDN can be reused.
3. Response to comment: Figure 1 doesn't help the reader to understand how the proposed mechanism works. Probably some information can be removed. A sequence diagram presenting the main tasks should be considered.
Thanks to the reviewers, we have replaced Figure 1 in the manuscript with a sequence diagram.
Figure 1. Flow of the NDN access control system
4. Response to comment: The authors should also start giving an overview of the main steps of the mechanism.
The main steps of the mechanism are divided as follows. The first step is for the CA to initialize the system to issue a globally unique identity to the AA and to issue a globally unique identity to the content consumer. The second step AA is responsible for managing the attributes of the attribute domain and generating the attribute public key. In addition, the AA will generate a proxy key and add it to the NDN routing node, and the AA will send the private key to the corresponding consumer user. In the third step, the publisher needs to use the public key to encrypt the information to be published. The fourth step is that the consumer user requests a key from the CA or uses the proxy key of the NDN node for partial decryption.
5. Response to comment: How the proposed mechanism avoids, the collusion attacks must be better explained.
In traditional attribute encryption schemes, a single AA combines parts of a user's private key (corresponding to different attributes) through key randomization. This randomization makes the elements of different keys of one user compatible with each other, but cannot be combined with the key elements of other users. This key randomization technique is not suitable for decentralized attribute encryption, because no CA can combine these elements, and each key comes from a different AA. In order to overcome this difficulty, this article uses different global identities to combine different user key parts to meet the needs of anti-collusion. Consumer users base this solution on the CP-ABE access control method to defend against conspiracy attacks. Suppose the number of permissions involved in ciphertext is n, and the number of collusion permissions is m. If , intuitively, these permissions can get all the keys that can be used to decrypt the content of ciphertext. If , there is at least one privilege that the key cannot be obtained. Therefore, ciphertext cannot be decrypted. This scheme implements a collusion attack of up to (n-1) privilege.
We tried our best to improve the manuscript and made some changes in the manuscript. These changes will not influence the content and framework of the paper.
We appreciate for Editors/Reviewers’ warm work earnestly, and hope that the correction will meet with approval.
Once again, thank you very much for your comments and suggestions.

Reviewer 2 Report
In this paper, the multi-authorization revocable access control method based on CP-ABE implements secure and effective access control of data in the NDN network. Due to the partial decryption on the NDN node, the decryption burden of the consumer client in the solution is reduced, and the method implements effective user and attribute revocation, and ensures forward security and backward security against collusion attacks. Finally, the performance and security analysis of the scheme of this paper proves that the scheme is safe and efficient.
Comments:
1 Give measurement of an encryption algorithm key size.
2 Compare the key size of other closely related algorithms.
3 Give more references in Table 2.
4 Add a section that clearly stands out the authors contribution
Author Response
Dear Editor of the Journal of Future Internet:
Thank you for your professional review and hard work! Thank you for your valuable comments and Suggestions! These Suggestions and opinions on our paper writing and research work are of great guiding significance! At the same time, we need to urge more serious attitude in scientific research and learning!
According to the opinions of the reviewers, we replied below.
Responds to the reviewer’s comments:
To Reviewer 2:
1. Response to comment: Give measurement of an encryption algorithm key size.
Response:
According to the comments of the reviewers, we give a measure of the key size of this scheme.
Let IA denote the universe set of all AA and each AA manages xk () attributes. For a user with uid, let denote the number of attributes obtained from the authority with .
2. Response to comment: Compare the key size of other closely related algorithms.
Response:
According to the reviewer's opinion, we compared the key size mentioned in the recent related literature with this article, and we inserted the result of the comparison into Table 2 in Question 3. The following is the result of our performance comparison between our scheme and the two schemes.
Figure 5. Relation between consumer decryption and encryption time and number of authorities
Figure 6. Relation between consumer decryption time and number of authorities
Figure 7. Relation between proxy decryption time and number of authorities
3. Response to comment: Give more references in Table 2.
Response:
According to the reviewers’ comments, we added two methods to Table 2 for comparison. The comparison results are shown below.
Table 2. calculation cost comparison
|
Program |
Secret key |
Ciphertext size |
Consumer encryption time |
Consumer decryption time |
|
Yang [15] |
||||
|
Qian [22] |
||||
|
Wu [23] |
||||
|
This scheme |
4. Response to comment: Add a section that clearly stands out the authors contribution
Response:
According to the reviewers' comments, we add the contribution of our scheme to the end of the first section.
Our contributions can be summarized as follows.
We propose a CP-ABE-based multi-authority revocable access control method. This method constructs a proxy-assisted access control scheme to achieve secure and effective access control of data in the NDN network. We design an effective user and attribute revocation method, and ensure forward security and backward security to prevent collusion attacks. In addition, from a comparative analysis of performance and safety, we show that our scheme is safer and more efficient than other schemes.
We tried our best to improve the manuscript and made some changes in the manuscript. These changes will not influence the content and framework of the paper.
We appreciate for Editors/Reviewers’ warm work earnestly, and hope that the correction will meet with approval.
Once again, thank you very much for your comments and suggestions.
